# Collateral positives of COVID-19 for culturally and linguistically diverse communities in Western Sydney, Australia

**Samuel Cornell**[1]*, **Julie Ayre**[1], **Olivia Mac**[1], **Raveena Kapoor**[1], **Kristen Pickles**[1], **Carys Batcup**[1], **Hankiz Dolan**[1], **Carissa Bonner**[1], **Erin Cvejic**[1], **Dana Mouwad**[2], **Dipti Zacharia**[2], **Una Tularic**[3], **Yvonne Santalucia**[4], **Ting Ting Chen**[2], **Gordana Basic**[2], **Kirsten McCaffery**[1], **Danielle Muscat**[1]

**1** Sydney Health Literacy Lab, School of Public Health, Faculty of Medicine and Health, The University of Sydney, Sydney, New South Wales, Australia, **2** Western Sydney Local Health District, North Parramatta, New South Wales, Australia, **3** Nepean Blue Mountains Local Health District, Kingswood, New South Wales, Australia, **4** Southwestern Sydney Local Health District, Warwick Farm, New South Wales, Australia

☉ These authors contributed equally to this work.
* Samuel.cornell@sydney.edu.au

**Data Availability Statement:** All relevant data are within the manuscript and its Supporting Information files.

## Abstract

### Background

To investigate whether culturally and linguistically diverse (CALD) communities in Western Sydney have experienced any positive effects during the COVID-19 pandemic, and if so, what these were.

### Methods

A cross–sectional survey with ten language groups was conducted from 21st March to 9th July 2021 in Sydney, Australia. Participants were recruited through bilingual multicultural health staff and health care interpreter service staff and answered a question, 'In your life, have you experienced any positive effects from the COVID-19 pandemic?' Differences were explored by demographic variables. Free–text responses were thematically coded using the Content Analysis method.

### Results

707 people completed the survey, aged 18 to >70, 49% males and 51% females. Only 161 (23%) of those surveyed reported any positive impacts. There were significant differences in the proportion of those who reported positives based on age (p = 0.004), gender (p = 0.013), language (p = 0.003), health literacy (p = 0.014), English language proficiency (p = 0.003), education (p = <0.001) and whether participants had children less than 18 years at home (p = 0.001). Content Analysis of open-ended responses showed that, of those that did report positives, the top themes were 'Family time' (44%), 'Improved self-care' (31%) and, 'Greater connection with others' (17%).

**Funding:** The authors received no specific funding for this work.

**Competing interests:** The authors declare that they have no competing interests.

## Discussion

Few surveyed participants reported finding any positives stemming from the COVID–19 pandemic. This finding is in stark contrast to related research in Australia with participants whose native language is English in which many more people experienced positives. The needs of people from CALD backgrounds must inform future responses to community crises to facilitate an equitable effect of any collateral positives that may arise.

## Introduction

The COVID-19 pandemic has impacted Australia since March 2020 and has been costly for the Australian population with widespread restrictions on movement and work between periods of control. Nevertheless, Australians have shown resilience [1]; they have identified positives as a side effect of restrictions [2] and generally complied well with public health directives; with high compliance rates reported by across multiple studies [3, 4].

The negative effects of the pandemic have been widely reported. However, research has found that people have adapted to the novel circumstances and often found positives amidst the disorder. Previous research, conducted in June 2020, with a national sample of Australians, found that 70% of participants had experienced positive effects of COVID-19; The three most common themes were 'Family time' (33%), 'Work flexibility' (29%) and 'Calmer life' (19%) [2]. Similarly, a study from Scotland conducted during weeks 9–12 of the Scottish lockdown from May to June 2020 found that participants reported feeling fitter, better rested and calmer—83% being more appreciative of things usually taken for granted, 67% more time doing enjoyable things, 62% paying more attention their health and 54% increasing their amount of exercise [5]. Furthermore, another Australian, qualitative, longitudinal survey found mixed responses from participants regarding the effects of COVID-19 on their family relationships—in which participants described feelings of loss and strains on relationships, but also of developing positive characteristics such as appreciation, gratitude, and tolerance and opportunities for strengthening family bonds [6].

Nevertheless, as with most aspects of health, previous research has not found an equitable distribution of positive experience, with those of higher socioeconomic status more likely to find positives, including working from home for pay and financial benefits [5]; while many existing inequities between the socioeconomic stratum have been exacerbated during COVID-19 [7, 8]. Furthermore, few studies have specifically aimed to ascertain the positive experience, if any, of those from culturally and linguistically diverse backgrounds. It is important to identify groups and populations which may not experience any positive effects arising from a disaster including a pandemic. This may be due to already present socioeconomic disparities which may be exacerbated from the detrimental effects of lockdowns and other pandemic related side–effects [9].

In this brief report, we present results from our survey conducted from March 21st to July 9th, 2021, survey which examined behaviour and intentions, information sources, and impacts of COVID–19 amongst people from culturally and linguistically diverse backgrounds in Greater Western Sydney. As part of the survey, we asked whether they had experienced any positive effects during the COVID–19 pandemic, and what those positive effects were.

## Methods

### Ethics approval

This study involves human participants and was approved by the Western Sydney Local Health District Human Research Ethics Committee (Project number 2020/ETH03085). Participants gave informed consent to participate in the study before taking part.

## Study design

This study involved a self–report cross-sectional survey with 11 language groups.

## Setting

Throughout the COVID-19 pandemic there has been numerous lockdowns and phases of restrictions affecting the residents of Sydney, New South Wales with concurrent widespread disruption to the daily lives of residents. The survey was conducted from March 21st to July 9th, 2021. During this period, the COVID–19 vaccines had begun to roll out across Australia, and daily cases in NSW ranged from 0 to 46 [10]. Stay at home orders (informally known as 'lockdown') were implemented across Greater Sydney on June 23rd [11]. On the day the survey closed (July 9th) the New South Wales (NSW) daily case count was 45.

Participants were recruited from Greater Western Sydney in NSW, Australia from three adjoining regions with high cultural diversity: Western Sydney, Southwestern Sydney, and Nepean Blue Mountains. According to data from PHIDU at Torrens University Australia, up to 39% of residents in these regions were born overseas in non-English speaking countries.

## Participants

Participants were eligible to take part if they were aged 18 years or over and spoke one of the following as their main language at home: Arabic, Assyrian, Chinese, Croatian, Dari, Dinka, Hindi, Khmer, Samoan, Tongan, Spanish. Through iterative discussions with Multicultural Health and Health Care Interpreter Service staff in each participating Local Health District, we selected 11 language groups that would provide broad coverage across different global regions, and groups with varying average levels of English language proficiency (based on 2016 Australian census data), varying access to translated materials and varying degrees of reading skill in their main language spoken at home.

## Recruitment

Participants were recruited through bilingual Multicultural Health staff and Health Care Interpreter Service staff. Multicultural Health staff recruited participants through their existing networks, community events and community champions. Health Care Interpreter Service staff recruited participants at the end of a medical appointment and via their community network. The survey was hosted online using the web-based survey platform Qualtrics. Potential participants were offered two means of taking part: completing the survey them- selves online (available in English or translated), or with assistance from bilingual staff or an interpreter who read the questions to them and recorded their responses. To ensure consistency in the phrases used for assisted survey completion, translated versions of the survey were provided to all staff assisting with survey completion. Translations were completed by translators with National Accreditation Authority for Translators and Interpreters (NAATI) accreditation where possible [12].

## Measures

Demographic survey items included age, gender, education, whether born in Australia, years living in Australia, main language spoken at home, English language proficiency, reading proficiency in language spoken at home, access to the internet, smartphones, chronic disease status, and a single-item health literacy screener 'How confident are you filling out medical forms by yourself?' [13]. The socioeconomic status of the area of residence for each individual was defined based on the SEIFA Index of Relative Socioeconomic Advantage and Disadvantage

(IRSAD [14]). IRSAD aligns the statistical local area with a decile ranking (1–10), with lower scores indicating greater socioeconomic disadvantage. The IRSAD decile was not available for some participants (n = 5), for example, because they had entered digits that did not correspond to a valid Australian postcode. IRSAD decile for these participants was replaced with the median IRSAD decile for speakers of the same language in the sample. For the analysis, IRSAD deciles were recoded into quintiles, and dichotomised (lowest quintile vs other).

Positive impacts of COVID–19 was assessed with a single-item, "In your life, have there been any positive effects from the COVID–19 pandemic?". Participants answered yes or no and could then provide free text feedback.

## Quantitative analysis

Quantitative data were analysed using IBM SPSS Statistics Version 26. Descriptive statistics were generated for demographic characteristics of the analysed sample. Within each language group, frequencies were weighted to reflect population (census data) gender and age group distributions (18–29 years, 30–49 years, 50–69 years, ≥70 years). A single participant indicated their gender as 'other' and was unable to be included in weighted analyses. Total recruitment for the Spanish language group were low (<50), with notable gaps for some age groups. For this reason, results for this language group are not presented in the analysis but are included in total frequencies. For the single item "positives", descriptive statistics were generated by age, gender, and health literacy, IRSAD and comorbidities. Chi-square tests were conducted to test for between-group differences in categorical variables. P values less than 0.05 were considered statistically significant. Descriptive statistics were also generated for "positives" by language group and free text responses were analysed via Content Analysis.

## Content analysis

Free-text responses to the item about positive impacts were analysed using Content Analysis [15], a widely used analysis method which combines qualitative and quantitative methods to analyse text data, allowing the content and frequency of categories to be reported. One member of the research team (KP) first read through all the valid free-text responses (n = 144) and developed the initial coding framework, based on a previously reported framework developed by SC [2], which was reviewed by the research team. 30 responses (~20%) were double coded independently by two members of the research team (OM and RK). Level of agreement was tested using Cohen's kappa (18) and indicated substantial agreement ($\kappa$ = 0.78). OM and RK then independently coded the remaining responses. The frequency of each code and main themes are reported.

## Results

### Sample characteristics

We had a total of 708 respondents. Sample characteristics are summarised in Table 1. The mean age was 45.4 years (standard error [SE] 0.78; range 18–91 years), and 51% of respondents were female (n = 363). Most participants (88%, n = 622) were born in a country other than Australia; 31% reported that they did not speak English well or at all (n = 220); 70% had no tertiary qualifications (n = 497). Inadequate health literacy was identified for 41% of the sample (n = 290).

### Positive impacts

Across the entire sample, only 23% of people reported that there had been any positive impacts of COVID–19 (n = 161). Number of people reporting positives by group is summarised in

**Table 1. Descriptive statistics of the total sample and those who reported positives.**

| Variable | Total number of sample | Total number reported positives with significance | |
|---|---|---|---|
| | N (%) | n (%) | P value# |
| **Age group** | | | **0.004** |
| 18–29 | 147 (20.7) | 28 (19.3) | |
| 30–49 | 295 (41.8) | 90 (30.5) | |
| 50–69 | 193 (27.3) | 34 (17.7) | |
| >70 | 72 (10.2) | 9 (12.5) | |
| **Gender*** | | | **0.013** |
| Male | 344 (48.6) | 61 (17.9) | |
| Female | 363 (51.4) | 100 (27.5) | |
| **Language** | | | **0.003** |
| Arabic | 80 (11.3) | 161 (22.8) | |
| Assyrian | 133 (18.8) | 25 (31.7) | |
| Chinese | 76 (17.1) | 25 (18.6) | |
| Croatian | 121 (6.2) | 11 (8.6) | |
| Dari | 44 (8.9) | 11 (24.4) | |
| Dinka | 63 (5.9) | 17 (26.8) | |
| Hindi | 42 (8.9) | 18 (42.0) | |
| Khmer | 63 (10.7) | 7 (10.5) | |
| Spanish** | 43 (5.9) | 25 (33.1) | |
| Samoan/Tongan | 42 (6.1) | 12 (28.3) | |
| **English language proficiency (How well do you speak English?)** | | | **0.003** |
| Very well/ well | 487 (68.9) | 127 (26.2) | |
| Not well/not at all | 220 (31.1) | 34 (15.4) | |
| **Literacy in a language other than English (How well do you read in your main language?)** | | | 0.774 |
| Very well/ well | 589 (83.4) | 136 (23.1) | |
| Not well/not at all | 118 (16.6) | 25 (21.6) | |
| **Health literacy*** | | | **0.014** |
| Adequate | 417 (58.9) | 110 (26.5) | |
| Inadequate | 290 (41.1) | 51 (17.6) | |
| **Education** | | | **<0.001** |
| Bachelor degree or above | 210 (29.7) | 77 (10.9) | |
| Below bachelor degree | 497 (70.3) | 84 (11.9) | |
| **Years living in Australia** | | | 0.778 |
| 5 years or less | 120 (16.9) | 31 (26.1) | |
| 6 to 10 years | 104 (14.7) | 20 (19.3) | |
| More than 10 years | 398 (56.4) | 92 (23.2) | |
| Born in Australia | 85 (12.0) | 18 (20.7) | |
| **IRSAD quintile** | | | 0.908 |
| Lowest | 224 (31.7) | 52 (23.1) | |
| Not lowest | 383 (68.3) | 109 (22.7) | |

*(Continued)*

**Table 1.** (Continued)

| Variable | Total number of sample | Total number reported positives with significance | |
|---|---|---|---|
| | N (%) | n (%) | P value# |
| **Self-reported chronic health conditions^** | | | 0.280 |
| 0 | 421 (59.6) | 106 (25.3) | |
| 1 | 154 (21.8) | 31 (20.2) | |
| 2 or more | 132 (18.6) | 24 (18.2) | |
| **Children less than 18 years** | | | <**0.001** |
| Yes | 262 (37.1) | 82 (31.4) | |
| No | 445 (62.9) | 79 (11.2) | |
| **Change in employment** | | | 0.316 |
| Yes | 273 (38.6) | 69 (25.3) | |
| No | 434 (61.4) | 92 (13.1) | |
| **Total** | 707 | 161 | |

Frequencies are weighted (using post-stratification weighting) to reflect each language group's gender and age group distribution (18–29 years, 30–49 years, 50–69 years, ≥70 years) based on 2016 census data for Western Sydney, South Western Sydney, and Nepean Blue Mountains' combined populations [17].

* 1 respondent indicated 'other/prefer not to say'.

** Spanish/Tongan numbers were too small and not included in analysis by language group.

*** Based on the Single Item Literacy Screener (SILS) [10].

^ Self-reported chronic health conditions included respiratory disease, asthma, chronic obstructive pulmonary disease, high blood pressure, cancer, heart disease, stroke, diabetes, depression, or anxiety.

# P value indicates the significance between groups based on chi-square tests.

Table 1. Reporting of positive impacts ranged from 12% (n = 9) for people aged seventy years or older to 30% (n = 90) for the 30–49-year age group. There were significant differences across language groups (p<0.001), the range was between 9% (n = 11) for Croatian speakers, to 42% (n = 18) for Hindi speakers. There were significant differences across genders; 18% (n = 61) of men reported positive impacts compared to 27.5% (n = 100) of women (p = 0.004), and 18% (n = 51) of people with inadequate health literacy reported positive impacts compared to 26% (n = 110) with adequate health literacy (p = 0.014). There was no significant difference among participants who did or did not report chronic health conditions when finding positives; 25% (n = 106) of people with no self-reported chronic health conditions reported positives compared to 20% (n = 31) with one and 18% (n = 24) with two or more self-reported chronic health conditions (p = 0.280). There was no significant difference between 23% (n = 52) of participants in the lowest IRSAD quintile reported positives compared to 23% (n = 109) not in the lowest. The proportion of people reporting positives was significantly higher for people with children less than 18 years (31%, n = 82) compared to those without (11%, n = 79; p = 0.001). There was no significant difference for those who reported positives between those who had experienced a change in employment status and those who had not (p = 0.316).

## Content analysis

Of the 161 participants who identified positive effects of COVID-19, 144 provided a written response detailing their positive experience(s).

**Table 2. Themes identified in free-text responses to question 'In your life, have you had any positive effects from the COVID-19 pandemic' with example quote, shown in decreasing order of frequency, of those who reported a positive.**

| Theme* | Example *quote* | N | % |
|---|---|---|---|
| More time with family | *"It made family members spend more time together" "More quality family time especially when we had a semi lockdown"* | 64 | 44.4 |
| Developed good habits and improved self-care | *"Pay more attention to developing good habits of personal hygiene, developing a healthy lifestyle" "...doing exercise and taking care of the diet"* | 45 | 31.3 |
| Greater connection with others | *"We're closer as a family, with co-workers and with the community" "I now cherish friendships and family relationships even more than I used to"* | 25 | 17.4 |
| Staying at home | *"Am a homebody so enjoyed not having to go out"* | 19 | 13.2 |
| Financial benefit | *"I was able to spend less money going out and socialising"* | 11 | 7.6 |
| Work flexibility | *"Working from home for some days"* | 11 | 7.6 |
| Increased time for hobbies and leisure | *"Learning to grow vegetables and flowers, learning to cook"* | 8 | 5.6 |
| Gained perspective | *"To value things, not allowing time to past not doing anything, reflect on what is important and what is not, be grateful for what I have and that I am not in need"* | 8 | 5.6 |
| Less time commuting | *"Working remotely saves on the time to be spent on the commute"* | 4 | 2.8 |
| Mental health improvement | *"Pay more attention to physical and mental health issues, increase the amount of exercise outdoor, pay more attention to people who need assistance, decline unnecessary social events, the betterment of one's inner-self / spirituality."* | 2 | 0.1 |

*Categories are not mutually exclusive.

The three most reported themes were as follows (Table 2):

1. 'Family time' (44.4%, n = 64), in which participants described the positive effects of being able to spend more time with their family (either online or within their home) and a feeling of greater appreciation for their loved ones.

2. 'Developed good habits and improved self-care' (31.3%, n = 45) with participants explaining that they had more time to give to their own wellbeing.

3. 'Greater connection with others' (17.4%, n = 25) with participants highlighting the time together during the pandemic had enabled a deeper connection with others in their community.

Other major themes in which more than 10% of participants identified positive effects included the following: (4) Staying at home; (5) Financial benefit, and (6) Work flexibility.

## Discussion

The findings of this paper illustrate the experiences of people from culturally and linguistically diverse communities living in greater Sydney, in which only 23% identified positive impacts stemming from the COVID–19 pandemic. Nevertheless, although fewer participants in the current study acknowledged positives compared to our previous research with a general Australian population in June 2020, the responses differed by frequency, but not by kind. There are strong similarities between the predominant themes of this study, contrasted to our previous work, in which the top themes were, 'family time', 'work flexibility' and 'calmer life'. It is notable that, in the current study, 44% of participants who identified a positive noted 'Family

time' [2]. This makes sense in the context of a crisis, when people become increasingly reliant on their family and community for support, and especially when under stay-at-home orders which necessitates spending more time with household members [16]. It is plausible that the current study sample found fewer positives, including of spending time with family, due to the likelihood of those in culturally and linguistically diverse communities having family ties overseas.

Our current research, juxtaposed with our previous study, provides an interesting lens into the experiences of those in culturally and linguistically diverse communities who reported far fewer positives—23% in the current study compared to 70% in our previous research [2]. It adds to a growing body of evidence which suggests that positive effects of COVID-19 have not been experienced equally. Our previous research highlighted that while some groups experienced positives stemming from the COVID-19 restrictions, particularly those living with others and working from home for pay, others did not and in fact were much more likely to experience adverse events such as becoming unemployed [17]. Additionally, research has expounded the issue that people of culturally and linguistically diverse groups, as well as women, the unemployed and those of poorer physical health are more likely to experience mental health issues during COVID–19 [18].

Although this research reports on the positives experienced by culturally and linguistically diverse participants, it is imperative to acknowledge that does not suggest the absence of negative effects. It is apparent that for many people in culturally and linguistically diverse communities, there have been many challenges attributable to the disruption of the pandemic which are highlighted in our parallel research. We found broad psychological, financial, and social impacts of the pandemic including significant numbers of respondents experiencing anxiety and worry, financial stress, and negative impacts on relationships; 25% of participants reported feeling nervous or stressed most or all of the time over the past week, 22% of participants reported feeling alone or lonely most or all of the time, 25% reported negative impacts on their relationships and 39% reported a change in employment status due to pandemic restrictions [19].

Clearly, the needs of the residents of culturally and linguistically diverse communities in Western Sydney need to be carefully considered by those in Government to ensure that those who are at greater risk of pandemic-related disadvantage are supported. It is prudent to foster greater community engagement, mental health services and economic / structural supports for these communities, with a focus on the linguistic and cultural barriers communities may face in a system not specifically designed with them in mind.

## Strengths and limitations

This study is novel in its use of both Content Analysis and quantitative analysis to determine if any positive outcomes are to be found in the experiences of a sample of culturally and linguistically diverse people resident in Sydney, New South Wales. It is the largest Australian survey which explores the impacts of COVID–19 for people who speak a language other than English at home.

However, it is important to consider that this investigation into "positives" experienced during the COVID-19 pandemic is dependent upon a single survey item administered during a specific and short time—March 21st to July 9th, 2021, when COVID-19 case numbers were low in Australia. Our previous research from June 2020 reports on a time in which most of Australia was leaving strict restrictions, but in our current study case numbers and restrictions were heading in the opposite direction; this may have been reflected in the much lower rate of positives found in this study amongst these communities.

It is unknown how the repercussions of lockdown, restrictions and higher risk of COVID-19 may have influenced culturally and linguistically diverse community members in terms of finding positives. It is possible that even fewer participants would have reported positives with continued lockdown, particularly as the communities which we surveyed faced tighter restrictions than the rest of Sydney as the weeks after the survey progressed, including curfews, limits on outdoor exercise [20] and the presence of the Australian Defence Force [21]. We are unable to explore changes in impacts over time in this study. Therefore, this brief report should be considered a starting point for further exploration of the themes identified related to positives and it should not be construed as a definitive source of evidence for the experiences of culturally and linguistically diverse residents in Western Sydney.

## Conclusion

Few participants reported finding any positives because of the COVID–19 pandemic and associated changes to daily life when surveyed between March 21st to July 9th, 2021. This finding is in stark contrast to related research conducted earlier in the pandemic in which many more people experienced positives. The needs of people from culturally and linguistically diverse backgrounds must be strongly considered in future crises responses to promote an equitable effect of any positives that may arise and importantly to negate any negative effects.

## Supporting information

**S1 File. CALD covid survey (OSF).**
(SAV)

**S2 File. CALD plan.**
(CSAPLAN)

## Acknowledgments

We would like to acknowledge and thank the community members who participated in this survey and the team who co-designed the survey, recruited the participants, and collected the data.

## Author Contributions

**Conceptualization:** Julie Ayre, Kristen Pickles, Hankiz Dolan, Carissa Bonner, Erin Cvejic, Dana Mouwad, Dipti Zacharia, Una Tularic, Yvonne Santalucia, Ting Ting Chen, Gordana Basic, Kirsten McCaffery, Danielle Muscat.

**Data curation:** Samuel Cornell, Julie Ayre, Olivia Mac, Kristen Pickles, Carissa Bonner, Kirsten McCaffery, Danielle Muscat.

**Formal analysis:** Samuel Cornell, Julie Ayre, Carissa Bonner, Erin Cvejic, Ting Ting Chen, Danielle Muscat.

**Funding acquisition:** Hankiz Dolan, Dana Mouwad, Dipti Zacharia, Una Tularic, Yvonne Santalucia, Ting Ting Chen, Danielle Muscat.

**Investigation:** Julie Ayre, Hankiz Dolan, Carissa Bonner, Erin Cvejic, Yvonne Santalucia, Gordana Basic, Kirsten McCaffery, Danielle Muscat.

**Methodology:** Julie Ayre, Kristen Pickles, Erin Cvejic, Kirsten McCaffery, Danielle Muscat.

**Project administration:** Olivia Mac, Raveena Kapoor, Carys Batcup.

**Supervision:** Danielle Muscat.

**Writing – original draft:** Samuel Cornell, Danielle Muscat.

**Writing – review & editing:** Samuel Cornell, Julie Ayre, Olivia Mac, Raveena Kapoor, Kristen Pickles, Carys Batcup, Hankiz Dolan, Carissa Bonner, Dana Mouwad, Dipti Zacharia, Una Tularic, Yvonne Santalucia, Ting Ting Chen, Gordana Basic, Kirsten McCaffery, Danielle Muscat.

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
