## [Decision Letter · Decision Letter 0]

26 Oct 2022

PONE-D-22-21636Collateral positives of COVID-19 for culturally and linguistically diverse communities  in Western Sydney, AustraliaPLOS ONE

Dear Dr. Cornell,

Thank you for submitting your manuscript to PLOS ONE. After careful consideration, we feel that it has merit but does not fully meet PLOS ONE’s publication criteria as it currently stands. Therefore, we invite you to submit a revised version of the manuscript that addresses the points raised during the review process.

Please address the issues raised by the Academic Editor when submitting your updated manuscript.

We look forward to receiving your revised manuscript.

Kind regards,

Joyce Addo-Atuah, PhD

Guest Editor

PLOS ONE

Journal Requirements:

"This study was not specifically funded. "

Additional Editors Comments:

Please address the concerns/questions below:

1) Full name of the Institutional Review Board (IRB) or the Ethics Committee that authorized the study

2) Outline the details of the recruitment of study participants here instead of referring potential readers to a prior publication

3) What is the single item health literacy screener used in this study?

Reviewers' comments:

Reviewer's Responses to Questions

**Comments to the Author**

1. Is the manuscript technically sound, and do the data support the conclusions?

Reviewer #1: Yes

2. Has the statistical analysis been performed appropriately and rigorously? 

Reviewer #1: Yes

3. Have the authors made all data underlying the findings in their manuscript fully available?

Reviewer #1: Yes

4. Is the manuscript presented in an intelligible fashion and written in standard English?

Reviewer #1: Yes

5. Review Comments to the Author

Reviewer #1: I would have liked to see greater depth to your survey and additional questions that sought to refute or further analyze the areas identified as positives from other studies. For example, what aspects of family time improved, further differentiation between self care aspects like diet or exercise or mental wellbeing. Is there a greater understanding of whether the relationship enhancement occurred more with family or with friends. When money was spent less on certain elements was it invested in different ways, was debts reduced? How was gratitude expressed to other? In paying attention to those in need, how was that quantified? I would liked to have seen greater emphasis of statistically significant categories and an attempt to hypothesis why findings were identified.

6. PLOS authors have the option to publish the peer review history of their article (what does this mean?). If published, this will include your full peer review and any attached files.

Reviewer #1: No

---

## [Author Response · Author response to Decision Letter 0]

1 Nov 2022

Please see response to editor and reviewer within the attached letter.

---

## [Editor Report · Decision Letter 1]

24 Nov 2022

Collateral positives of COVID-19 for culturally and linguistically diverse communities in Western Sydney, Australia

PONE-D-22-21636R1

Dear Dr. Cornell,

We’re pleased to inform you that your manuscript has been judged scientifically suitable for publication and will be formally accepted for publication once it meets all outstanding technical requirements.

Kind regards,

Joyce Addo-Atuah, PhD

Guest Editor

PLOS ONE

Additional Editor Comments (optional):

Good, updated manuscript has addressed all reviewer concerns
---

## [Editor Report · Acceptance letter]

1 Dec 2022

PONE-D-22-21636R1 

Collateral positives of COVID-19 for culturally and linguistically diverse communities in Western Sydney, Australia 

Dear Dr. Cornell:

I'm pleased to inform you that your manuscript has been deemed suitable for publication in PLOS ONE. Congratulations! Your manuscript is now with our production department. 

Kind regards, 

on behalf of

Dr. Joyce Addo-Atuah 

Guest Editor

PLOS ONE